# Koopman-Assisted Reinforcement Learning

**Preston Rozwood**$^{\alpha\beta*}$, **Edward Mehrez**$^{\alpha\beta*}$, **Ludger Paehler**$^{\delta}$, **Wen Sun**$^{\beta}$, **Steven L. Brunton**$^{\gamma}$

Subharmonic Technologies$^{\alpha}$, Cornell University$^{\beta}$,
Technical University of Munich$^{\delta}$, University of Washington$^{\gamma}$
{pwr36, ejm322}@cornell.edu, ludger.paehler@tum.de,
ws455@cornell.edu, sbrunton@uw.edu

## Abstract

The Bellman equation and its continuous form, the Hamilton-Jacobi-Bellman (HJB) equation, are ubiquitous in reinforcement learning (RL) and control theory contexts due, in part, to their guaranteed convergence towards a system's optimal value function. However, this approach has severe limitations. This paper explores the connection between the data-driven Koopman operator and Bellman Markov Decision Processes, resulting in the development of two new RL algorithms to address these limitations. In particular, we focus on Koopman operator methods that reformulate a nonlinear system by lifting into new coordinates where the dynamics become linear, and where HJB-based methods are more tractable. These transformations enable the estimation, prediction, and control of strongly nonlinear dynamics. Viewing the Bellman equation as a controlled dynamical system, the Koopman operator is able to capture the expectation of the time evolution of the value function in the given systems via linear dynamics in the lifted coordinates. By parameterizing the Koopman operator with the control actions, we construct a new "Koopman tensor" that facilitates the estimation of the optimal value function. Then, a transformation of Bellman's framework in terms of the Koopman tensor enables us to reformulate two max-entropy RL algorithms: soft-value iteration and soft actor-critic (SAC). This highly flexible framework can be used for deterministic or stochastic systems as well as for discrete or continuous-time dynamics. Finally, we show that these algorithms attain state-of-the-art (SOTA) performance with respect to traditional neural network-based SAC and linear quadratic regulator (LQR) baselines on three controlled dynamical systems: the Lorenz system, fluid flow past a cylinder, and a double-well potential with non-isotropic stochastic forcing. It does this all while maintaining an interpretability that shows how inputs tend to affect outputs, what we call "input-output" interpretability.

## 1 Introduction

Interpretability is frequently lost in RL algorithms, especially those driven by large neural networks. In this paper, we re-examine the underlying Bellman equation through a dynamical systems lens and a novel application of a fundamental *transfer operator* known as the **Koopman operator** [18, 19, 24, 3]. Nonlinear dynamics may be represented in terms of this infinite-dimensional linear Koopman operator, which acts on the space of all possible measurement functions of the system. Mathematically, given a function from the state space to the reals $g : \mathcal{X} \to \mathbb{R}$, the Koopman operator $\mathcal{K}$ for deterministic (time-homogenous) autonomous systems is defined as:

$$\mathcal{K}g(x) := g(F(x)) = g(x'), \tag{1}$$

where $F$ is the single-step flow map or law of motion. More generally, in stochastic autonomous systems, it is defined as the conditional forecast operator:

$$\mathcal{K}g(x) = \mathbb{E}\left(g(X')|X = x\right). \tag{2}$$

NeurIPS 2023 AI for Science Workshop.

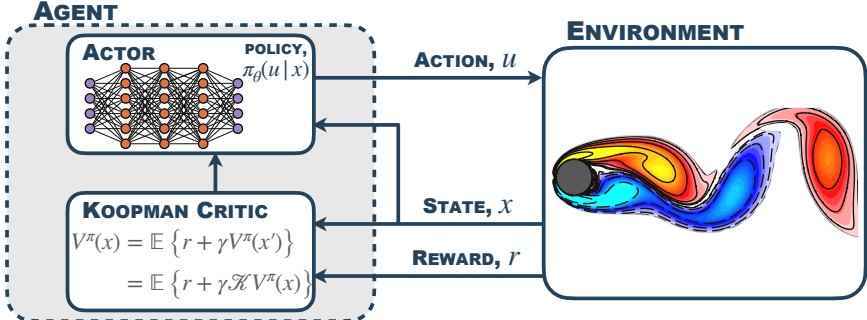

Figure 1: Koopman-assisted reinforcement learning in the example of the Soft Actor Koopman-Critic, a Koopman variant of the popular Soft Actor-Critic algorithm. The *Koopman Critic* receives the state, and the reward as original, nonlinear dynamics, before lifting these dynamics onto the vector space, where they can be advanced in time with the *Koopman operator* linearly. This critique is then fed back to the Actor which issues the action to be performed in the environment.

In this paper, we recast the continuation term in the Bellman equation in terms of the Koopman operator:

$$V^\pi(x) = \mathbb{E}_{u \sim \pi(\cdot|x)} \left\{ r(x,u) + \gamma \mathbb{E}_{x' \sim p(\cdot|x,u)} \left[ V(x') \right] \right\} \tag{3a}$$

$$\implies V_\pi(x) = \mathbb{E}_{u \sim \pi(\cdot|x)} \left\{ r(x,u) + \gamma \mathcal{K}^u V(x) \right\}. \tag{3b}$$

It is important to note that since the Koopman operator is equivalent to the conditional forecast operator given state $x$, it only relies on the current state. In addition, the Koopman operator's dependence on the action $u$ has been made explicit via the $\mathcal{K}^u$ notation.

The Koopman operator allows us to rewrite the equations of motion of a nonlinear system as a linear system of equations on an infinite-dimensional function space [24, 7, 23, 3, 4]. When considering a particular observable, like the value function in reinforcement learning, for many problems it is possible to isolate a finite set of basis, or dictionary, functions for which we can approximately express the dynamics of the value function with a finite-dimensional Koopman operator/matrix. Many such approaches have been explored in traditional applied dynamical systems such as dynamic mode decomposition (DMD) [30] for linear or approximately linear systems, and extended dynamic mode decomposition (EDMD) [8], as well as the closely related sparse identification of nonlinear dynamics (SINDy) [5], which can be thought of as generalizing the previous approaches to linear mappings rather than just linear operators of dynamics [4]. This body of work was subsequently extended to stochastic dynamical systems [16, 17] and controlled dynamical systems [15, 28].

An important contribution of this work is the construction of the **Koopman tensor** in a multiplicatively separable dictionary space on states and controls, respectively. We then solve for this tensor via least squares for the prediction of future state dictionaries using (state, control, future state) samples collected from simulated environments. Slices of the resulting Koopman tensor are then weighted by the control dictionary elements to construct a finite-dimensional Koopman matrix in the lifted state observable space for any control value, including those outside of the training dataset.

Finally, as a proof of concept of *Koopman assisted RL algorithms*, we reformulate two max-entropy RL algorithms: **soft value iteration** [13] and **soft actor-critic** [11, 12]. We refer to this approach broadly as **Koopman-Assisted Reinforcement Learning (KARL)** and to the two particular reformulated algorithms as **soft Koopman value iteration (SKVI)** and **soft actor Koopman critic (SAKC)**. We validate these algorithms in four environments using quadratic cost functions: linear system, the Lorenz system, fluid flow past a cylinder, and a double-well potential with non-isotropic stochastic forcing. We demonstrate in each system that our KARL methods achieve SOTA or near-SOTA compared to traditional neural network-based SAC and LQR baselines.

## 1.1 Related Work

Use of the Koopman operator in RL is an emerging field of research. Previous works have used the Koopman operator for imitation learning and identifying symmetries in state dynamics [31, 33]. Koopman analysis has been extended to controlled systems [28, 15]; however, our main contribution

of recasting Markov Decision Processes (MDPs) using Koopman embedding frameworks has not been explored. Notably, we introduce a novel approach to parameterize the Koopman operator using a Koopman tensor on a lifted state-control space which is essential for handling controlled dynamical systems. Other attempts to incorporate Koopman into controlled systems include dynamic mode decompositon with control (DMDc) [28] for system identification and applications of LQR to Koopman-linearized dynamics [28, 15]. The Koopman operator has also been used for model predictive control (MPC) [6, 20, 21, 29], although without the same convergence guarantees as KARL. One recent work learns a Koopman autoencoder [22, 25, 27] for Q-learning [33]. In contrast, our approach of rewriting the Bellman and HJB equations using control-dependent Koopman operators enables both interpretability and single-step estimates of the expectation of the value function.

## 2   Koopman-Assisted Reinforcement Learning (KARL)

### 2.1   Technical Background

Here, we discuss relevant theory and algorithms. We review Koopman operator theory, discuss the reformulation of MDPs, and build intuition for the Koopman tensor. Finally we apply insights from our reformulation of MDPs to develop two new max entropy RL algorithms: **soft Koopman value iteration (SKVI)** and **soft actor Koopman critic (SAKC)**.

#### 2.1.1   Koopman Operator Theory

The Koopman operator describes the time evolution for any function of the state of autonomous and controlled systems. Formally, we consider real-valued vector measurement functions $g \colon M \to \mathbb{R}$, which are themselves elements of an infinite-dimensional Hilbert space and where $M$ is a manifold that represents the state space. Typically the observables are assumed to belong to the function space $L^\infty(\mathcal{X})$ where $M = \mathcal{X} \subset \mathbb{R}^d$ . Often, these functions $g$ are called **observables** or **measurements** of the state $x$. Formally, the **Koopman operator** $\mathcal{K} \colon L^\infty(\mathcal{X}) \to L^\infty(\mathcal{X})$ for deterministic (time-homogenous) autonomous systems is defined as:

$$\mathcal{K}g(x) := g(F(x)) = g(x'), \tag{4}$$

where $F$ is the single-step flow map or law of motion. More generally in stochastic autonomous systems, it is defined as the conditional forecast operator:

$$\mathcal{K}g(x) = \mathbb{E}\left(g(X')|X_t = x\right). \tag{5}$$

**Remark 2.1 (Basis Functions of the Koopman Operator)** *The main insight from this operator theoretic view of dynamics is that a finite set of basis functions may be found to characterize observables of the state as long as the **Koopman operator has a finite point spectra rather than a continuum**. This perspective is crucial below as we lift our coordinate spaces to a finite set of basis functions, typically called **dictionary functions**, which we denote by $\{\phi_i\}$. See A.1 (brief technical exposition); [23] (broader treatment of Koopman operator in applied dynamical systems).*

### 2.2   MDPs and Bellman's Equation

Below, we assume an infinite horizon Markov decision process (MDP) setting for the agent's objective. In discrete time, assuming that the agent follows policy $\pi$ which is a distribution over actions, the $\pi$-value function takes the form:

$$V^\pi(x) = \mathbb{E}\left[\sum_{t=0}^{\infty} -\gamma^t c(x_t, u_t)\,\middle|\, \pi, x_0 = x\right], \tag{6}$$

where $\gamma \in [0, 1]$ represents the discount rate and we express rewards in terms of negative costs $r(x, u) = -c(x, u)$ as we will use a quadratic cost function as in the standard setting LQR below.

**Remark 2.2 (Finite-Horizon MDPs and the Time-Inhomogenous Koopman Operator)** *The finite horizon MDP can also be transformed using the Koopman operator, however, in that case, the operator will not be time-homogeneous and will depend not only on action, but also on point in time. For an excellent in-depth discussion of discrete-time MDPs (both finite and infinite horizon) and their place in RL, see [1].*

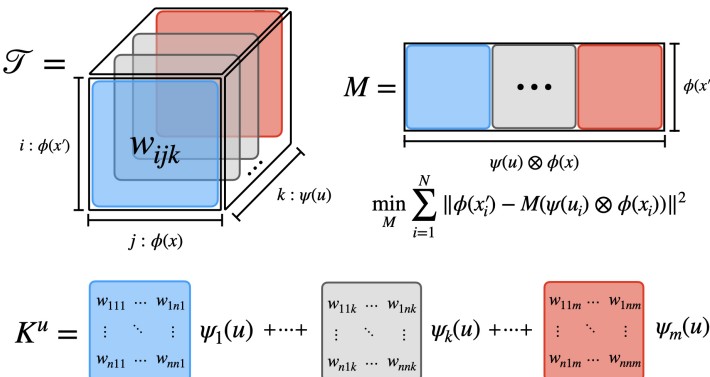

Figure 2: Construction of action-dependent Koopman operators $K^u$ from the Koopman tensor $K$. Colors match along the $k$ index (depth of tensor box). Each of the matrix slices is then weighted according to the $\psi$ dictionary elements to construct the control-dependent Koopman operator $K^u$.

The agent's optimal value function is expressed as follows:

$$V^*(x) = \max_\pi V^\pi(x). \tag{7}$$

We use the Bellman equation (or Hamilton-Jacobi-Bellman equation in continuous time) to recursively characterize the above optimal value function. The discrete Bellman optimality equation is:

$$V(x) = \max_{\pi_t} \mathbb{E}_{u \sim \pi(\cdot|x)} \left[ -c(x, u) + \gamma \mathbb{E}_{x' \sim p(\cdot|x, u)}[V(x')] \right] \tag{8a}$$

$$= \max_\pi \mathbb{E}_{u \sim \pi_t(\cdot|x)} \left[ -c(x, u) + \gamma \mathcal{K}^u V(x) \right] \tag{8b}$$

$$= \max_\pi \mathbb{E}_{u \sim \pi_t(\cdot|x)} \left[ -c(x, u) + \gamma w^T K^u \phi(x) \right], \tag{8c}$$

where $\phi : \mathcal{X} \to \mathbb{R}^{d_x}$ is the dictionary vector function $\phi = (\phi_1, \cdots, \phi_{d_x})$ that serves as a basis for the Koopman operator applied to the value function. Note that the Koopman operator applies element-wise when applied to vector functions.

## 2.3 Koopman Tensor Formulation of Controlled Dynamics

We introduce the Koopman tensor as an extended Koopman operator that includes a third dimension specifically to capture control action observables. With it, we are able to construct a 2D Koopman operator from any action on a continuum by collapsing the action observable dimension.

---

**Algorithm 1** Koopman Tensor Estimation

---

**Require:** State feature map $\phi : \mathcal{X} \to \mathbb{R}^{d_x}$, control feature map $\psi : \mathcal{U} \to \mathbb{R}^{d_u}$, and a sample $\{(x_i, u_i)\}_{i=0}^N$
 1: Solve for $\widehat{M}$ as in Equation 9
 2: Convert $M$ into $K$ tensor using Fortran-style reshaping

---

Denote $\phi : \mathcal{X} \mapsto \mathbb{R}^{d_x}$ as the feature mapping (each coordinate of $\phi$ is an observable function), and $\psi : \mathcal{U} \mapsto \mathbb{R}^{d_u}$ as the feature mapping for control. Let us seek a finite-dimensional approximation of the Koopman operator $\mathcal{K}^{u_i}$. Denote $K \in \mathbb{R}^{d_x \times d_x \times d_u}$ as a 3D tensor as shown in Figure 2. For any $u$, let us denote $K^u \in \mathbb{R}^{d_x \times d_x}$ as follows: $K^u[i, j] = \sum_{z=1}^{d} K(i, j, z)\psi(u)[z]$. Namely, $K^u$ is the result of the tensor vector product along the third dimension of $K$ and $K^u$ serves as the finite-dimensional approximation of Koopman operator $\mathcal{K}^u$. We learn $K$ as follows:

$$\min_K \sum_{i=1}^N \left\| K^{u_i} \phi(x_i) - \phi(x'_i) \right\|^2.$$

We can slightly rewrite the above objective so that it becomes the regular multi-variate linear regression problem. We can rearrange to write $K$ as a 2-dimensional matrix in $\mathbb{R}^{d_x \times d_x \cdot d_u}$. Denote $M \in \mathbb{R}^{d_x \times d_x \cdot d_u}$, where $M[i,:] \in \mathbb{R}^{d_x \cdot d_u}$ is the vector from stacking the columns of the 2D matrix $K[i,:,:]$. A nice way to visualize this rearrangement can be seen in the definition of M in Figure 2. Denote $\psi(u) \otimes \phi(x) \in \mathbb{R}^{d_x \cdot d_u}$ as the Kronecker product. Thus we have:

$$K^u \phi(x) = M(\psi(u) \otimes \phi(x)).$$

Therefore, the optimization problem becomes a regular linear regression:

$$\min_M \sum_{i=1}^N \|M(\psi(u) \otimes \phi(x_i)) - \phi(x_i')\|^2. \tag{9}$$

Once we compute $M$, we can convert back to Koopman operator for any $u \in \mathcal{U}$ by reshaping $M$ back to the 3D tensor $K$. Then the $d_x \times d_x$ finite dimensional Koopman operator approximation is again $K^u$ for any $u \in \mathcal{U}$ as seen in the summation in Figure 2.

Note that the above formulation also works for discrete control set $\mathcal{U}$. The benefit of doing this is that control share information and similarity via their feature $\psi(u)$. This could give better sample efficiency than learning independent Koopman operators one for each discrete control.

## 2.4 Max Entropy Koopman RL Algorithms

To demonstrate the effectiveness of how the Koopman operator can be used in RL we follow a popular strand of RL literature and add an **entropy penalty** $\alpha \ln \pi$ to the cost function to encourage exploration of the environment.

### 2.4.1 Koopman Value Iteration

In addition to the assumption about the finite dimensional representation of the Koopman operator, we will also assume that the optimal value function $V^\star(x)$ can be written as a linear combination of basis functions. In other words, there exists a $w^\star \in \mathbb{R}^d$, such that $V^\star(x) = (w^\star)^\top \phi(x)$. Given a $w \in \mathbb{R}^d$, we can express the (entropy regularized) Bellman error as follows. For any $x$:

$$w^\top \phi(x) - \min_{\pi:\mathcal{X} \mapsto \Delta(\mathcal{U})} \left[ \mathbb{E}_{u \sim \pi(x)} \left[ c(x,u) + \alpha \ln \pi(u|x) + w^\top K^u \phi(x) \right] \right].$$

Thanks to the entropy regularization, given a $w$, we can express the optimal form of $\pi$ as follows:

$$\pi(u|x) = \exp\left( -\left( c(x,u) + w^\top K^u \phi(x) \right) / \alpha \right) / Z_x, \tag{10}$$

where $Z_x$ is the normalizing constant that is only dependent on $x$ that makes $\pi(\cdot|x)$ a proper probability distribution. Note that $\pi$ depends on $w$.

Converting this into an iterative procedure to find the value function weights, $w'$ in terms of the previous weights $w$, the **average Bellman error (ABE)** over the dataset can be expressed as follows:

$$\min_{w':\|w\|_2 \leq W} \frac{1}{N} \sum_{i=1}^N \left( w'^\top \phi(x) - \min_{\pi(\cdot|x)} \left[ \mathbb{E}_{u \sim \pi(x)} \left[ c(x,u) + \alpha \ln \pi(u|x) + w^\top K^u \phi(x) \right] \right] \right)^2. \tag{11}$$

The above is a canonical ordinary least squares (OLS) problem and can thus be solved explicitly. We repeat this procedure until the ABE is small or until there is minimal improvement between update steps. Unfortunately, given finite data, it is possible for a suboptimal value function to satisfy the Bellman equation exactly [10]. As a result, we need to be careful when using the Bellman error as a training objective for RL agents.

## 2.5 Soft Actor Koopman-Critic (SAKC)

Here, we outline how we modify the the Soft-Actor-Critic (SAC) framework [11] to restrict the search space by incorporating information from the Koopman operator. Using the same loss functions and similar notation to that of the SAC paper [11], we first specify the soft value function loss:

$$J_V(w) = \mathbb{E}_{x \sim \mathcal{D}} \left[ \frac{1}{2} \left( V_w(x) - \mathbb{E}_{u \sim \pi_\phi} \left[ Q_\theta(x,u) - \alpha \ln \pi_\nu(u|x) \right] \right)^2 \right]. \tag{12}$$

**Algorithm 2** Learning Optimal Policy via Average Bellman Error (ABE) Minimization

---

**Require:** Confidence parameter $\epsilon$, reward function or reward function approximator $r$, and feature maps on the state $\phi : \mathcal{X} \to \mathbb{R}^{d_x}$ and the control $\psi : \mathcal{U} \to \mathbb{R}^{d_u}$. Two datasets $\mathcal{D}_x$ and $\mathcal{D}_x$ that store a comprehensive set of possible states of the environment. Initialize $w_0$ as desired.

1: Let $ABE$ denote the objective of the minimization problem (11).
2: Let $\pi_0^*(u|x)$ be the optimal policy given in (10) evaluated at $w_0$
3: $ABE(w_t, \pi_t) = ABE(w_0, \pi_0^*)$
4: **while** $ABE(w_t, \pi_t) > \epsilon$ **do**
    Sample $x_i \sim_{iid} \mathcal{D}_x$
    Compute costs, log probabilities, and value function outputs of all $(x_i, u)$ pairs for all $u \in \mathcal{D}_u$
    $w_{t+1} =$ solution to Equation (11)
    $w_{t+1} \mapsto \pi_{t+1}^*(u|x)$
    $ABE(w_t, \pi_t) = ABE(w_{t+1}, \pi_{t+1}^*(u|x))$
5: **end while**

---

The additional specification that is imposed in the Koopman RL framework would be a restriction around the specifications of $V_w(x)$ and $Q_\theta(x, u)$:

$$V_w(x) = w^T \phi(x), \tag{13}$$

where $w$ is a vector of coefficients for the dictionary functions.

Next, we show how the loss function for the (Q) quality function changes:

$$J_Q(\theta) = \mathbb{E}_{(x,u)\sim\mathcal{D}} \left[ \frac{1}{2} \left( Q_\theta(x, u) - \widehat{Q}(x, u) \right) \right], \tag{14}$$

where the target Q-function incorporates the Koopman operator and is defined as:

$$\begin{aligned} \widehat{Q}(x, u) &= r(x, u) + \gamma \mathbb{E}_{x'\sim p(\cdot|x,u)} \left[ V_{\bar{w}}(x') \right] \\ &= r(x, u) + \gamma \bar{w}^T K^u \phi(x), \end{aligned} \tag{15}$$

where $\mathcal{K}$ represents the infinite-dimensional Koopman for a fixed action $u$ and $K^u$ represents the Koopman operator's finite-dimensional form on the state-dictionary space.

Finally, the loss function for the policy does not change and is given by:

$$J_\pi(\nu) = \mathbb{E}_{x\sim\mathcal{D}} \left[ D_{KL} \left( \pi_\nu(\cdot|x) \left\| \frac{\exp(Q_\theta(x, \cdot))}{Z_\theta(x)} \right. \right) \right]. \tag{16}$$

After these adjustments, the general algorithm remains the same as in the SAC paper and is given by:

---

**Algorithm 3** Soft Actor Koopman-Critic

---

**Require:** Initial parameter vectors $w, \bar{w}, \theta, \nu$

1: **for** each iteration **do**
2:     **for** each environment step **do**
        $u \sim \pi_\nu(u|x)$
        $x' \sim p(x'|x, u)$
        $\mathcal{D} \leftarrow \mathcal{D} \cup \{(x, u, r(x, u), x'\}$
3:     **end for**
4:     **for** each gradient step **do**
        $w \leftarrow w - \lambda_V \hat{\nabla}_\nu J_V(w)$
        $\theta_i \leftarrow \theta_i - \lambda_Q \hat{\nabla}_{\theta_i} J_Q(\theta_i)$ for $i \in \{1, 2\}$
        $\nu \leftarrow \nu - \lambda_\pi \hat{\nabla}_\nu J_\pi(\nu)$
        $\bar{w} \leftarrow \tau w + (1 - \tau)\bar{w}$
5:     **end for**
6: **end for**

---

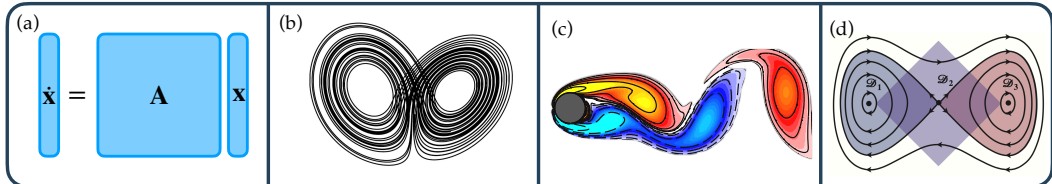

Figure 3: Four benchmark problems investigated: (a) simple linear system; (b) Lorenz 1963 model; (c) incompressible fluid flow past a cylinder [at Reynolds number 100]; and (d) double-well potential with non-isotropic stochastic forcing.

## 3 Experimental Evaluation

### 3.1 Evaluation Design

We apply KARL in a linear system, fluid flow past a cylinder, Lorenz, and a double-well potential: figure 3 for an illustration of the dynamics for the four systems. For details such as formal definitions and variable values of these systems, please refer to A.2. For purposes of evaluating the proposed algorithms, we build upon CleanRL [14] from which we use the reference implementations of the corresponding RL algorithms,[1] and upon which our implementation of Soft Actor Koopman-Critic is based.[2] For the Koopman tensor, we use an order 2 monomial dictionary space for all systems, and allow the Koopman tensor access to 30,000 steps of data collected by a random agent for each environment. Each environment has different action ranges, which were selected with some minimal a priori understanding of the system, but are mostly random. To ensure that we train the various policies to minimize the same cost, we use the LQR cost function for each system, defined as:

$$c(x, u) = x^T Q x + u^T R u. \tag{17}$$

### 3.2 Results

Evaluating the performance of the Soft Actor Koopman-Critic (SAKC) on the four environments against the classical control baseline, as well as the reference soft actor-critic implementations, we see that on the *Linear System* the SAKC needs $\approx 5,000$ environment steps to properly calibrate its Koopman critic, and match the SOTA performance of LQR, Value Iteration, and the value-based SAC. The Q-function based SAC exhibits a slightly greater degree of instability and requires $\approx 8,000$ environment steps to match SOTA performance. On the *Fluid Flow* this training dynamic is matched with SAKC requiring $5,000 - 8,000$ environment steps to calibrate itself in this more difficult environment and reach SOTA. The required number of environment steps matches those required by the value-function based SAC. On the even more difficult chaotic *Lorenz System* SAKC requires slightly more exploration with $10,000$ environment steps, before hitting SOTA. Of note here is that SAKC exhibits more performance instability as compared to the better-performing SAC, and Value Iteration. On the *Double Well*, we see a fast calibration of SAKC to quickly match SOTA, while LQR is unable to match the performance of the reinforcement learning algorithms on this even more difficult environment. Zooming in on the performance of the different RL algorithms 5, we can see that after seemingly reaching SOTA, the episodic returns of all agents equally fluctuate and SAKC matches the performance of the other SAC methods.

When tasked with learning dynamics and converging to an optimal controller in non-linear systems, our new Koopman-assisted reinforcement learning approach significantly outperformed LQR in non-linear systems, as measured by relative cost. See Figure 4 and Table 1. KARL's success in these unique dynamical settings demonstrates its significant flexibility across varying environments: fluid flow shows its prowess in non-linear systems; Lorenz proves KARL's ability to control chaotic systems; and double well shows its ability to learn in stochastic environments.

## 4 Conclusion and Future Work

In summary, we present a novel approach to RL by integrating Koopman operator methods with existing maximum entropy RL algorithms to formulate Koopman-Assisted Reinforcement Learning.

---

[1] https://github.com/vwxyzjn/cleanrl
[2] https://github.com/Pdbz199/koopman-rl

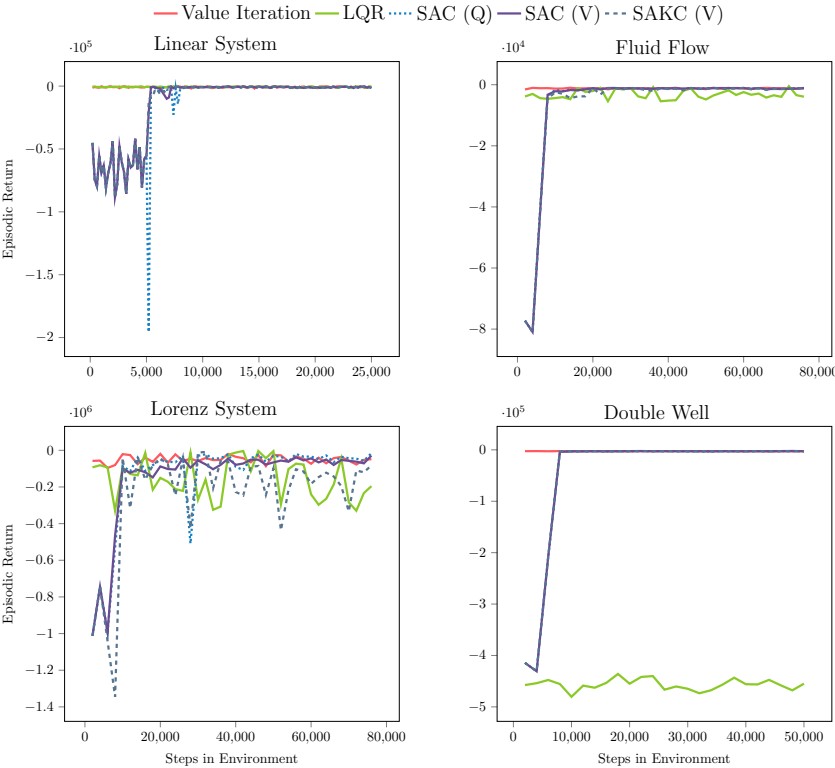

Figure 4: Episodic returns of the evaluation environments for the compared algorithms: value iteration, a linear quadratic regulator (LQR), the Q-function based soft actor-critic (SAC (Q)), the V-function based soft actor-critic (SAC (V)), and the soft actor Koopman-critic (SAKC).

By leveraging the Koopman operator, KARL overcomes limitations of traditional RL algorithms by making them more "input-output" interpretable, allowing for a deeper understanding of the learned policies and their underlying dynamics. The empirical results presented in this paper on a diverse set of dynamical systems-based environments including non-linear, chaotic, and stochastic systems show that KARL is able to match the SOTA performance of the reference Soft Actor-Critic algorithm. In addition, KARL outperforms the classical control baseline of LQR methods on the non-linear environments, showcasing its flexibility, and adaptability. KARL's SOTA performance in these various complex systems highlights its potential for real-world applications, making it a valuable addition to the repertoire of current RL techniques. The future of KARL lies in its continuous evolution and adaptation to more complex and realistic settings. Addressing these challenges and exploring these directions will allow the Koopman operator to aid in the development of robust, interpretable, and efficient future RL algorithms.

Prospects for further development and application of KARL are both numerous and promising. For example, integration of KARL with modern online learning techniques [32] could support real-time applications, especially in combination with techniques to improve the efficiency of the algorithm such as knowledge gradients [9]. A comprehensive theoretical analysis of KARL algorithms, including convergence properties and sample complexity bounds, would provide valuable insights into their behavior, and would aid in providing more intuition and guarantees for safety-critical applications. Incorporating sparsification techniques such as SINDy [5] to facilitate the use of large dictionary spaces in unfamiliar complex systems to determine value function dynamics will help further interpret the main driving features (observables) of the optimal value function. Visualization techniques to better interpret the learned Koopman tensor and Koopman-dependent operators could facilitate broader adoption in domains where interpretability is a top priority, such as healthcare, economics, and autonomous driving systems.

## Acknowledgements

The authors would like to thank Stefan Klus who contributed code for the Double-well system, edits to old drafts, and many insightful conversations about continuous-time stochastic dynamical systems. SLB acknowledges support from the National Science Foundation AI Institute in Dynamic Systems (grant number 2112085) and from the Army Research Office (ARO W911NF-19-1-0045).

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

# Appendix

## A  Further Theoretical Background

### A.1  Koopman Operator Theory Details

In what follows, we discuss more technical details of both theory and algorithms. The Koopman operator describes the time evolution for any function of the state of autonomous and controlled systems. Formally, we consider real-valued vector measurement functions $g \colon M \to \mathbb{R}$, which are themselves elements of an infinite-dimensional Hilbert space and where $M$ is a manifold. Typically this manifold is taken to be $L^{\infty}(\mathcal{X})$ where $\mathcal{X} \subset \mathbb{R}^d$. Often, these functions $g$ are called observables. In deterministic autonomous systems, we denote the law of motion of the state $x$ as:

$$x' = \mathbf{F}(x), \tag{18}$$

for discrete-time systems and

$$\dot{x}(t) = \mathbf{f}(x(t)), \tag{19}$$

for continuous-time systems.
Similarly, in deterministic controlled systems, we can denote the law of motion as:

$$x' = \mathbf{F}(x, u), \tag{20}$$

for discrete-time systems and

$$\dot{x}(t) = \mathbf{f}(x(t), u(t)), \tag{21}$$

for continuous-time systems.

For discrete-time stochastic systems we can express the (controlled) law of motion as

$$x' = \mathbf{F}(x, u) + \sigma(x, u)\varepsilon' \tag{22}$$

where $\{\varepsilon\}$ is a white noise process. For continuous-time stochastic systems, we will only give as an example the stochastic differential equation form for Itô-diffusion processes, however the Koopman operator theory applies more broadly to any Markov process:

$$dX = \mu(X, u)dt + \sigma(X, u)dW \tag{23}$$

where $W$ is a Wiener process (i.e. standard Brownian motion).

**Remark A.1** *One can obtain a discrete-time dynamical system by applying a transformation to a continuous-time dynamical. This transformation can be thought of as sampling the system at discrete points in time so that $x_k = x(k\Delta t)$, where $\Delta t$ is the time step. This transformation is called a flow map, or discrete-time propagator, and, in the deterministic case, for example, is given by*

$$x(t + \Delta) = \mathbf{F}_{\Delta t}(x(t)) = x(t) + \int_t^{t+\Delta} \mathbf{f}(x(\tau))d\tau.$$

The Koopman operator $\mathcal{K}$, and its (infinitesimal) generator $\mathcal{L}$, is an infinite-dimensional linear operator that acts on observables $g$ as:

$$\mathcal{K}g = g \circ F \tag{24}$$

$$\mathcal{L}g = f \cdot \nabla g. \tag{25}$$

The Koopman generator $\mathcal{L}$ has the following limiting relationship with the Koopman operator:

$$\mathcal{L}g = \lim_{t \to 0} \frac{\mathcal{K}g - g}{t} = \lim_{t \to 0} \frac{g \circ F - g}{t}. \tag{26}$$

In stochastic systems, the definition of the Koopman operator is generalized to be defined as:

$$\mathcal{K}g = \mathbb{E}(g(X)|X_0 = \cdot)$$

$$\mathcal{L}g = \lim_{t \to 0} \frac{\mathcal{K}g - g}{t},$$

where $\{X\}$ denotes the stochastic process representing the state over time.

Next, we briefly discuss how to represent the Koopman operator on a set of basis functions. Consider a function space $\mathcal{G} \subset \mathcal{X} \mapsto \mathbb{R}$. For any $u \in \mathcal{U}$, the Koopman operator $\mathcal{K}^u$ associated with $u$ is defined as follows:

$$\mathcal{K}^u g := g \circ F(\cdot, u) = \mathbb{E}_{x' \sim F(\cdot, u)}[g(x')].$$

### A.1.1 Continuous-time MDPs and Bellman's Equation

In continuous time, the value function from following policy $\pi$ takes the form:

$$V^\pi(x) = \mathbb{E}\left[\left. \int_{t=0}^{\infty} -e^{-\rho t} c(x(t), u(t)) dt \, \right| \pi, x(0) = x \right]. \tag{27}$$

The agent's optimal value function is again defined as:

$$V^*(x) = \max_\pi V^\pi(x). \tag{28}$$

We now turn to the Hamilton-Jacobi-Bellman to recursively characterize the above optimal value function.

$$V(x) = \frac{1}{\rho} \max_\pi \mathbb{E}_{u \sim \pi(\cdot|x)}\left[ -c(x, u) + \mathcal{L}^u V(x) \right]. \tag{29}$$

where $\mathcal{L}$ is the generator of the Koopman operator as defined above.

### A.1.2 Koopman Tensor for Discrete and Continuous-time Dynamics

As discussed above in Section 2.3, the Koopman tensor is constructed by finding the best finite dimensional operator (matrix) such that given the dictionary spaces on the action and state space, we can best predict the next observable value, one step ahead in time, in an OLS sense. As a result, the slices of the tensor as depicted in Figure 2 when applied to the dictionary on actions evaluated at a specific action leads to the Koopman operator for that specific action. In a deterministic setting, the Koopman operator for a fixed action when applied to the state dictionary space evaluated at a fixed state represents a characterization of the dynamics of the system going from a starting state $x$ to a future state $x'$. In a stochastic setting, the Koopman operator gives the conditional expectation of $x'$.

The Koopman tensor can be extended to support continuous-time dynamics using the Koopman generator operator. The minimization is as follows: $\sum_i \| \frac{d}{dt} \phi(x_i) - M(\psi(u_i) \otimes \phi(x_i)) \|$ and the $M$ can be reshaped into the Koopman tensor as shown in Figure 2.

## A.2 System Dynamics

Here we provide details about the four dynamical systems used to compare various RL algorithms in the results.

### A.2.1 Linear System

The equations for a linear system are given by

$$F(x, u) = Ax + Bu \tag{30}$$

where $A$ and $B$ are matrices.

### A.2.2 Fluid Flow

To approximate the fluid flow past a circular cylinder, we use the following reduced-order model developed by Noack et al. [26]:

$$f(x, u) = \begin{bmatrix} \mu x_0 - \omega x_1 + A x_0 x_2 \\ \omega x_0 + \mu x_1 + A x_1 x_2 + u \\ -\lambda(x_2 - x_0^2 - x_1^2) \end{bmatrix} \tag{31}$$

where $\mu = 0.1$, $\omega = 1.0$, $A = -0.1$, and $\lambda = 1$. The states $x_0$ and $x_1$ represent the most energetic proper orthogonal decomposition modes for the flow, and the third state $x_2$ represents the shift mode [26], which is important for capturing relevant transients. This model has been used to test Koopman modeling algorithms in the past [22].

### A.2.3 Lorenz

The Lorenz system is a canonical benchmark dynamical system given by the equations

$$f(x, u) = \begin{bmatrix} \sigma(x_1 - x_0) + u \\ (\rho - x_2)x_0 - x_1 \\ x_0x_1 - \beta x_2 \end{bmatrix} \tag{32}$$

where $\sigma = 10$, $\rho = 28$, and $\beta = \frac{8}{3}$. This system can be challenging for Koopman-based approaches because it has a continuous eigenvalue spectrum [2].

### A.2.4 Stochastic Double Well Potential

For a stochastic test system, we consider the stochastically forced particle in a double well potential, governed by the following dynamics:

$$f(x, u) = \begin{bmatrix} 4x_0 - 4x_0^3 + u \\ -2x_1 + u \end{bmatrix} + \begin{bmatrix} 0.7 & x_0 \\ 0 & 0.5 \end{bmatrix} \begin{bmatrix} v_0 \sim \mathcal{N}(0, 1) \\ v_1 \sim \mathcal{N}(0, 1) \end{bmatrix}. \tag{33}$$

## B More Details on Results

### B.1 Linear Quadratic Regulator Implementation

For our implementation of LQR, we added a file to our fork of the CleanRL repo and made use of the Python Control Systems Library. Specifically, we use the $dlqr$ function for the linear system and the $lqr$ function for controlling the rest of the systems. The linear system is the only system that has inherent $\mathbf{A}$ and $\mathbf{B}$ matrices as in $\mathbf{A}x + \mathbf{B}u$, so to recover usable $\mathbf{A}$ and $\mathbf{B}$ matrices for the other three systems, we linearize the system dynamics around whichever fixed (reference) point we are looking to control the system towards. For computing cost, the Qs and Rs are identity matrices for all systems except in Lorenz where the R matrix is an identity matrix multiplied by $0.001$ to incentivize more action from the agent.

### B.2 Discrete Value Iteration Implementation

In our fork of the CleanRL GitHub repository, we have added a new algorithm file for discrete value iteration. We manually implemented the calculation of the average Bellman error given a set of system states, and the extraction of the softmax policy. We use the same dataset to train our value iteration policy as the Koopman tensor. For our discretized action space, we use the min and max action in the system's range and choose some somewhat arbitrary number of actions into which we can evenly split the range. After the value iteration policy is trained, we ran it against the same initial conditions as the other algorithms as shown in Figure 4.

### B.3 Continuous Value Iteration Implementation

Because there is an inherent, undesirable limitation when working with a discrete action space, we attempted a simple method of continuous value iteration that simply used the trained value function weights $w$ from the discrete value iteration above and trained a continuous policy neural network to minimize the KL-divergence between the values from $c(x_i, u_j) + \alpha \ln \pi(u_j|x_i) + \gamma w^T K^{u_j}(x_i)$ and the log probabilities from the policy model.

### B.4 Average Cost Comparisons for SKVI

Comparing the cost of the application of RL with discrete value iteration to the application to the classical control approach of LQR we witness the following difference in computing costs:

### B.5 Detailed Return Comparisons for Double Well Environment

| Average Cost Over 100 Episodes | | | |
|---|---|---|---|
| | **LQR** | **Discrete Value Iteration** | **Reduction in Average Cost** |
| **Linear System** | **690.3652** | 746.2580 | -8.1% |
| **Fluid Flow** | 1,609.8472 | **1,528.9954** | 5.0% |
| **Lorenz** | 1,475,234.6126 | **160,554.6264** | 89.1% |
| **Double Well** | 103,704.5169 | **2,661.4114** | 97.4% |

Table 1: Illustrates the comparative average cost incurred by LQR and value iteration policies over 100 episodes. Bolded values indicate the lowest average cost, and thus, the best performing controller for each dynamical system. The last column shows how the KARL value iteration controller compared to the LQR controller expressed as a percentage reduction in average cost.

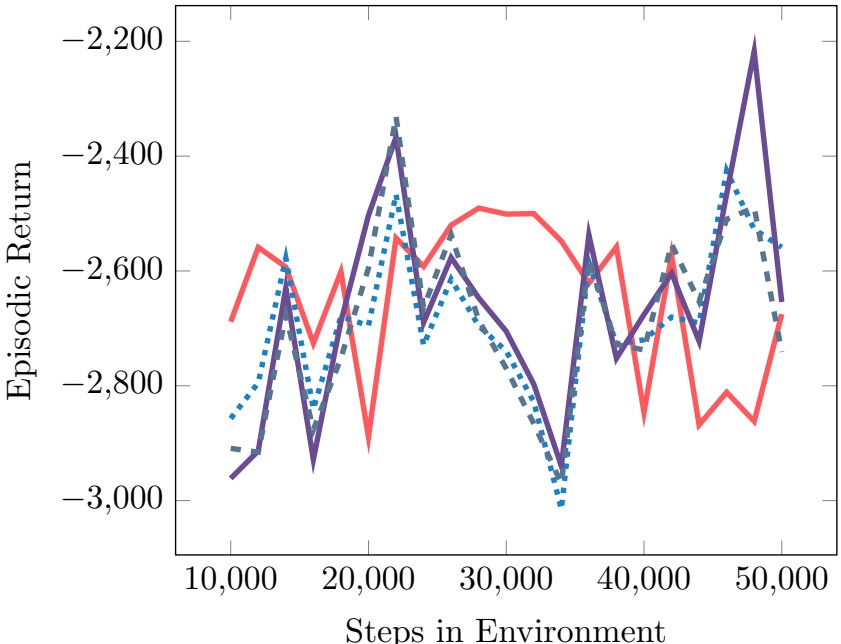

Figure 5: Zoomed in episodic returns for the double well, where the linear quadratic regulator has been discarded due to its significantly worse performance as compared to the reinforcement learning algorithms.

