# OpenReview forum: "Koopman-Assisted Reinforcement Learning"
_NeurIPS.cc/2023/Workshop/AI4Science — NeurIPS2023-AI4Science Oral_

### Official Review · Reviewer_whGe · 2023-10-25
**review of reviewer whGe**

**Rating:** 7
**Confidence:** 3

**Review:**

This paper explores how the Koopman operator, a data-driven tool, connects with the Bellman equation and the Hamilton-Jacobi-Bellman (HJB) equation used in reinforcement learning and control theory. The aim is to overcome limitations in their application. The Koopman operator helps transform nonlinear systems into a new coordinate system with linear dynamics, making it easier to work with HJB-based methods. This transformation enables the estimation and control of highly nonlinear dynamics. The paper introduces a "Koopman tensor" to estimate the optimal value function. By using the Koopman operator to describe the value function's time evolution and introducing control actions, two new reinforcement learning algorithms are developed: soft-value iteration and soft actor-critic (SAC). These algorithms outperform traditional approaches in various systems.

## pros
1. Present a novel integration between model-free RL and Koopman theory
2. Increase the interpretability while maintaining the similar performance in the empirical results

## cons
1. It should be better to demonstrate more visualization analysis to investigate the explainability of the results.

---

### Meta-Review · Area_Chair_b1VP · 2023-10-26

**Recommendation:** Accept (Oral)
**Confidence:** 5

**Metareview:**

The idea of incorporating Koopman operator into Reinforcement Learning is novel and the paper is well-written. The resulting performance is convincing. The paper is recommended for acceptance.